# A Deep-Learning-Based CPR Action Standardization Method

**DOI:** 10.3390/s24154813

**Published:** 2024-07-24

**Authors:** Yongyuan Li, Mingjie Yin, Wenxiang Wu, Jiahuan Lu, Shangdong Liu, Yimu Ji

**Affiliations:** 1Jiangsu Tuoyou Information Intelligent Technology Research Institute Co., Ltd., Nanjing 210012, China; lyy573303122@163.com; 2School of Computer Science, Nanjing University of Posts and Telecommunications, Nanjing 210023, China; 1023040815@njupt.edu.cn (M.Y.); 1223045119@njupt.edu.cn (W.W.); lsd@njupt.edu.cn (S.L.); 3School of Internet of Things, Nanjing University of Posts and Telecommunications, Nanjing 210023, China; 2023070801@njupt.edu.cn

**Keywords:** deep learning, processing speed, cardiopulmonary resuscitation, defibrillators, reference standards, posture

## Abstract

In emergency situations, ensuring standardized cardiopulmonary resuscitation (CPR) actions is crucial. However, current automated external defibrillators (AEDs) lack methods to determine whether CPR actions are performed correctly, leading to inconsistent CPR quality. To address this issue, we introduce a novel method called deep-learning-based CPR action standardization (DLCAS). This method involves three parts. First, it detects correct posture using OpenPose to recognize skeletal points. Second, it identifies a marker wristband with our CPR-Detection algorithm and measures compression depth, count, and frequency using a depth algorithm. Finally, we optimize the algorithm for edge devices to enhance real-time processing speed. Extensive experiments on our custom dataset have shown that the CPR-Detection algorithm achieves a mAP0.5 of 97.04%, while reducing parameters to 0.20 M and FLOPs to 132.15 K. In a complete CPR operation procedure, the depth measurement solution achieves an accuracy of 90% with a margin of error less than 1 cm, while the count and frequency measurements achieve 98% accuracy with a margin of error less than two counts. Our method meets the real-time requirements in medical scenarios, and the processing speed on edge devices has increased from 8 fps to 25 fps.

## 1. Introduction

Out-of-hospital cardiac arrest (OHCA) is a critical medical emergency with a substantial impact on public health, exhibiting annual incidence rates of approximately 55 per 100,000 people in North America and 59 per 100,000 in Asia. Without timely intervention, OHCA can lead to irreversible death within 10 min [1]. Studies have demonstrated that CPR and AED defibrillation performed by nearby volunteers or citizens significantly improve survival rates [1,2,3]. Standard CPR procedures are known to enhance survival outcomes in cardiac arrest patients [3]. However, the dissemination of CPR skills remains limited in many countries, primarily relying on mannequins and instructors, leading to high costs and inefficiencies. Traditional AED devices also lack the capability to prevent harm caused by improper operation [4,5].

Current CPR methods have several limitations, particularly in their effectiveness during real emergency situations. Traditional CPR training relies heavily on classroom simulations, which cannot replicate the pressure and urgency of actual cardiac arrest scenarios. This can lead to improper performance during real emergencies [6]. Although virtual reality (VR) and augmented reality (AR) technologies are being used to enhance CPR training, they remain primarily educational tools and are not widely integrated into real-time emergency applications [7,8]. Moreover, mainstream CPR techniques have not fully incorporated artificial intelligence (AI) assistance; advancements have focused more on mechanical devices and VR/AR training rather than real-time AI intervention [8,9]. Recent advancements in CPR algorithms have started to address these issues by integrating computational models and machine-learning techniques. For instance, the use of integrated computational models of the cardiopulmonary system to evaluate current CPR guidelines has shown potential in improving CPR effectiveness [10]. Additionally, machine learning has been used to identify higher survival rates during extracorporeal cardiopulmonary resuscitation, significantly enhancing survival outcomes [11]. The future trends in CPR technology indicate that the combination of AI and machine learning will continue to evolve, potentially predicting and shaping technological innovations in this field [12]. To bridge the gap between training and real-time application, this paper proposes the first application of posture-estimation and object-detection algorithms on AEDs to assist in real-time CPR action standardization, extending their use to actual emergency rescues. This innovative approach addresses the lack of real-time AI-assisted intervention in current CPR methods, thereby improving the accuracy and effectiveness of lifesaving measures during OHCA incidents. By integrating AI technology into AED devices, we aim to provide immediate feedback and corrective actions during CPR, potentially increasing survival rates and reducing risks associated with improper CPR techniques. This approach represents a significant advancement over traditional methods, which lack the ability to dynamically adjust in real-time and guide rescuers [13,14].

To enhance real-time medical interventions, advanced pose estimation techniques like OpenPose are highly beneficial. Developed by the Perceptual Computing Lab at Carnegie Mellon University, OpenPose is a pioneering open-source library for real-time multi-person pose estimation. It detects human body, hand, facial, and foot keypoints simultaneously [15]. Initially, OpenPose used a dual-branch CNN architecture to produce confidence maps and part affinity fields (PAFs) for associating body parts into a coherent skeletal structure. Subsequent improvements focused on refining PAFs, integrating foot keypoint detection, and introducing multi-stage CNNs for iterative prediction refinement [16,17]. Supported by continuous research and updates, OpenPose remains robust and efficient for edge computing and real-time applications [18], solidifying its status as a leading tool in diverse and complex scenarios.

In addition, deploying neural-network models on AED edge devices to recognize and standardize rescuers’ CPR actions can effectively improve the survival rate of cardiac arrest patients. However, deploying neural-network models on embedded systems faces challenges, such as high weight, insufficient computational power, and low running speed [19]. Most early lightweight object detection models were based on MobileNet-SSD (single shot multibox detector) [20]. Installing these models on some high-end smartphones can achieve sufficiently high running speeds [21]. However, due to insufficient ARM cores for running neural networks, model execution speed is slow on low-cost advanced RISC machine (ARM) devices [22].

In recent years, various lightweight object-detection networks have been proposed and widely applied in traffic management [23,24,25,26], fire warning systems [27], anomaly detection [28,29,30], and facial recognition [31,32,33]. Redmon et al. [34] introduced an end-to-end object-detection model using Darknet53, incorporating k-means clustering for anchor boxes, multi-label classification for class probabilities, and a feature pyramid network for multi-scale bounding box prediction. Wong et al. [35] developed Yolo Nano, a compact network for embedded object detection with a model size of approximately 4.0 MB. Hu et al. [36] improved the Yolov3-tiny network by using depthwise distributed convolutions and squeeze-and-excitation blocks, creating Micro-Yolo to reduce parameters and optimize performance. Lyu [37] proposed NanoDet, an anchor-free model using generalized focal loss and GhostPAN for enhanced feature fusion, increasing accuracy on the COCO dataset by 7% mAP. Ge et al. [38] modified Yolo to an anchor-free mode with a decoupled head and SimOTA strategy, significantly enhancing performance. For example, Yolo Nano achieved 25.3% AP on the COCO dataset with only 0.91 M parameters and 1.08 G FLOPs, surpassing NanoDet by 1.8%, while the improved Yolov3 AP increased to 47.3%, exceeding the current best practice by 3.0%. Yolov5 Lite [39] optimized inference speed by adding shuffle channels and pruning head channels while maintaining high accuracy. Dogqiuqiu [40] developed the Yolo-Fastest series for single-core real-time inference, reducing CPU usage. Yolo-FastestV2 used the ShufflenetV2 backbone, decoupled the detection head, reduced parameters, and improved the anchor-matching mechanism. Dogqiuqiu [41] further proposed FastestDet, simplifying to a single detection head, transitioning to anchor-free, and increasing candidate objects across grids for ARM platforms. However, for our dataset, FastestDet underperformed, mainly due to its single detection head design, limiting the utilization of features with different receptive fields and lacking sufficient feature fusion, resulting in insufficient accuracy in locating small objects.

This paper proposes a standardized CPR action-detection method based on AED, utilizing skeletal points to assist in posture estimation. We develop the CPR-Detection algorithm based on Yolo-FastestV2, which includes a novel compression depth-calculation method that maps actual depth by analyzing the wristband’s displacement. Additionally, we optimize the computation for edge devices to enhance their speed and accuracy. The main contributions of this paper include:(1)Introducing a novel method called deep-learning-based CPR action standardization (DLCAS) and developing a custom CPR action dataset. Additionally, we incorporated OpenPose for pose estimation of rescuers.(2)Proposing an object-detection model called CPR-Detection and introducing various methods to optimize its structure. Based on this, we developed a new method for measuring compression depth by analyzing wristband displacement data.(3)Proposing an optimized deployment method for automated external defibrillator (AED) edge devices. This method addresses the issues of long model inference time and low accuracy that exist in current edge device deployments of deep-learning algorithms.(4)Conducting extensive experimental validation to confirm the effectiveness of the improved algorithm and the feasibility of the compression depth-measurement scheme.

## 2. Methods

As shown in Figure 1, the overall workflow of this study is divided into three parts. The first part involves the experimental preparation phase, which includes dataset collection, image pre-processing and augmentation, dataset splitting, training, and then testing the trained model to obtain performance metrics. The second part presents the flowchart of the DLCAS, covering pose estimation, object-detection network, and depth measurement, ultimately yielding depth, compression count, and frequency. The third part describes the model’s inference and application. The captured images, processed through the optimized AED edge devices, eventually become CPR images with easily assessable metrics.

In this section, we first introduce the principles of OpenPose, followed by the design details of CPR-Detection. Next, we explain the depth measurement scheme based on object-detection algorithms. Finally, we discuss the optimization of computational methods for edge devices.

### 2.1. OpenPose

In edge computing devices for medical posture assessment, processing speed and real-time performance are crucial. Therefore, we chose OpenPose for skeletal-point detection due to its efficiency and accuracy. OpenPose employs a dual-branch architecture that generates confidence maps for body-part detection and part affinity fields (PAFs) to assemble these parts into a coherent skeletal structure. This method enables precise and real-time posture analysis, which is essential for medical applications. Traditional pose-estimation algorithms often involve complex computations that delay processing. OpenPose optimizes this process by focusing on key points and their connections, significantly reducing computational load and improving speed. It detects body parts independently before associating them, enhancing accuracy and efficiency by minimizing redundant computations. Overall, OpenPose allows for accurate and swift identification and assessment of human postures, making it ideal for real-time medical applications. Its efficient processing and reduced computational overhead make it suitable for deployment in edge computing devices used in emergency medical care, ensuring both reliability and speed in critical situations.

As shown in Figure 2, the workflow of OpenPose starts with feature extraction through a backbone network. These features pass through Stage 0, producing keypoint heatmaps and PAFs. Keypoint heatmaps indicate confidence scores for the presence of keypoints at each location, while PAFs encode the associations between pairs of keypoints, capturing spatial relationships between different body parts. These outputs are refined in subsequent stages, iteratively improving accuracy. Finally, the keypoint heatmaps and PAFs are processed to generate the final skeletal structure, combining keypoints according to the PAFs to form a coherent and accurate representation of the human pose. This method ensures precise and real-time posture analysis, making it highly effective for applications in medical posture assessment, particularly in edge computing devices used in emergency medical care, ensuring both reliability and speed in critical situations [16].

### 2.2. CPR-Detection

In this study, we provide a detailed explanation of CPR-Detection. As illustrated in Figure 3, the model consists of three components: the backbone network ShuffleNetV2, the STD-FPN feature-fusion module, and the detection head. The STD-FPN feature-fusion module incorporates the MLCA attention mechanism, and the detection head integrates PConv position-enhanced convolution.

#### 2.2.1. PConv

In edge computing devices for medical emergency care, we need to prioritize processing speed due to performance and real-time processing requirements. Therefore, we chose partial convolution (PConv) to replace depthwise separable convolution (DWSConv) in Yolo-FastestV2. PConv offers higher efficiency while maintaining performance, meeting the needs for real-time processing [42].

As shown in Figure 4a, DWSConv works by first performing depthwise convolution on the input feature map, grouping by channels, and then using 1 × 1 convolution to integrate all channel information. However, this depthwise convolution can lead to computational redundancy in practical applications. The principle of PConv, illustrated in Figure 4b, involves performing regular convolution operations on a portion of the input channels while leaving the other channels unchanged. This design significantly reduces computational load and memory access requirements because it processes only a subset of feature channels. PConv only performs convolution on a specific proportion of the input features, resulting in lower FLOPs compared to DWSConv, thereby reducing computational overhead and improving model efficiency. In summary, PConv enhances the network’s feature representation capability by focusing on crucial spatial information without sacrificing detection performance.

This strategy not only improves the network’s processing speed but also enhances the extraction and focus on key feature channels, making it essential for real-time object-detection systems. Additionally, by reducing redundant computations, the application of PConv lowers model complexity and increases model generalization, ensuring robustness and efficiency in complex medical emergency scenarios. Therefore, PConv is an ideal convolution method for medical emergency devices, enabling real-time object detection while ensuring reliability and efficiency on edge computing devices.

#### 2.2.2. MLCA

In emergency medical scenarios, complex backgrounds can interfere with the effective detection of wristbands. To address this, we introduce the mixed local channel attention (MLCA) module to enhance the model’s performance in processing channel-level and spatial-level information. As illustrated in Figure 5, MLCA combines local and global context information to improve the network’s feature representation capabilities. This focus on critical features enhances both the accuracy and efficiency of target detection [43].

The core of MLCA lies in its ability to process and integrate both local and global feature information simultaneously. Specifically, MLCA first performs two types of pooling operations on the input feature vector: local pooling, which captures fine-grained spatial details, and global pooling, which extracts broader contextual information. These pooled features are then sent to separate branches for detailed analysis. Each branch output is further processed by convolutional layers to extract cross-channel interaction information. Finally, the pooled features are restored to their original resolution through an unpooling operation and fused using an addition operation, achieving comprehensive attention modulation. Compared to traditional attention mechanisms, such as SENet [44] or CBAM [45], MLCA offers the advantage of considering both global dependencies and local feature sensitivity. This is particularly important for accurately locating small-sized targets. Moreover, the design of MLCA emphasizes computational efficiency. Despite introducing a complex context fusion strategy, its implementation ensures that it does not significantly increase the network’s computational burden, making it well-suited for integration into resource-constrained edge devices. In performance evaluations, MLCA demonstrates significant advantages. Experimental results show that models incorporating MLCA achieve a notable percentage increase in mAP0.5 compared to the original models while maintaining low computational complexity.

Overall, MLCA is an efficient and practical attention module ideal for target detection tasks in emergency medical scenarios requiring high accuracy and real-time processing.

#### 2.2.3. STD-FPN

In recent years, ShuffleNetV2 [46] has emerged as a leading network for lightweight feature extraction, incorporating innovative channel split and channel shuffle designs that significantly reduce computational load and the number of parameters while maintaining high accuracy. Compared to its predecessor, ShuffleNetV1 [47], ShuffleNetV2 demonstrates greater efficiency and scalability, with substantial innovations and improvements in its structural design and complexity management. The network is divided into three main stages, each containing multiple ShuffleV2Blocks. Data first passes through an initial convolution layer and a max pooling layer, progressively moving through the stages, and ultimately outputs feature maps of three different dimensions. The entire network optimizes feature extraction performance by minimizing memory access.

As shown in Figure 6a, the FPN structure of Yolo-FastestV2 utilizes the feature map from the third ShuffleV2Block in ShuffleNetV2, combined with 1×1 convolution to predict large objects. These feature maps are then upsampled and fused with the feature maps from the second ShuffleV2Block to predict smaller objects. However, Yolo-FastestV2’s FPN only uses two layers of shallow feature maps, limiting the acquisition of rich positional information and affecting the semantic information extraction and precise localization of small objects. Considering that AED devices are typically placed within 50 cm to 75 cm from the patient, and the wristband is a small-scale target, we propose an improved FPN structure named STD-FPN (see Figure 6b), which effectively merges shallow and deep feature maps from ShuffleV2Block, focusing on small-object detection. Each output from the ShuffleV2Block is defined as $S_{i}$, i∈[1,3]. After processing through the MLCA module, it becomes $C_{i}$. First, $C_{1}$ is globally pooled to reduce its size by a factor of four to get $C_{1}^{\prime }$, which is then concatenated with $C_{3}$. This concatenated feature undergoes Convolution-BatchNormalization-ReLU(CBR), forming the input for the first detection head. The second detection head, designed for small objects, processes $C_{2}$ through CBR operations to match the channel count of $C_{1}$ and then upsamples $C_{2}^{\prime }$ along all dimensions using a specified scaling factor. $C_{2}^{\prime }$ is element-wise added to $C_{1}$, followed by the CBR operation.

After each feature-fusion step, a 1×1 convolution is applied. During the entire model training process, convolution helps extract effective features from previous feature maps and reduces the impact of noise. By using additive feature fusion, shallow and deep features are fully integrated, producing fused feature maps rich in object positional information, thus enhancing the original model’s localization capability.

### 2.3. Depth Measurement Method

Image processing often involves four coordinate systems: the world coordinate system, the camera coordinate system, the image coordinate system, and the pixel coordinate system. Typically, the transformation process starts from the world coordinate system, passes through the camera coordinate system and the image coordinate system, and finally reaches the pixel coordinate system [48]. Assume world coordinate point $P_{w}={\left(x_{w},y_{w},z_{w}\right)}^{T}$, camera coordinate point $P_{c}={\left(x_{c},y_{c},z_{c}\right)}^{T}$, image coordinate point $m={\left(x_{p},y_{p},1\right)}^{T}$, and pixel coordinate point $Pix={\left(u,v,1\right)}^{T}$. The transformation from the world coordinate point $P_{w}={\left(x_{w},y_{w},z_{w}\right)}^{T}$ to the camera coordinate point $P_{c}={\left(x_{c},y_{c},z_{c}\right)}^{T}$ is given by Formula (1).
(1)$$\left[\begin{array}{c} x_{c}\\ y_{c}\\ z_{c}\\ 1 \end{array}\right]=\left[\begin{array}{cc} R & T\\ \vec{0} & 1 \end{array}\right]\left[\begin{array}{c} x_{w}\\ y_{w}\\ z_{w}\\ 1 \end{array}\right]$$

In this formula, the orthogonal rotation matrix $R=\left[\begin{array}{ccc} r_{11} & r_{12} & r_{13}\\ r_{21} & r_{22} & r_{23}\\ r_{31} & r_{32} & r_{33} \end{array}\right]$ and the translation matrix $T={\left[\begin{array}{ccc} t_{x} & t_{y} & t_{z} \end{array}\right]}^{T}$. Assume the center O of the projective transformation as the origin of the camera coordinate system, and the distance from this point to the imaging plane is the focal length f. According to the principle of similar triangles, Formula (2) can be obtained to transform from the camera coordinate point $P_{c}={\left(x_{c},y_{c},z_{c}\right)}^{T}$ to the image coordinate point $m={\left(x_{p},y_{p},1\right)}^{T}$:(2)$$z_{c}m=\left[\begin{array}{cccc} f & 0 & 0 & 0\\ 0 & f & 0 & 0\\ 0 & 0 & 1 & 0 \end{array}\right]P_{c}$$

Assume that the length and width of a pixel are $d_{x}$, $d_{y}$, respectively. Pixel coordinate point $Pix={\left(u,v,1\right)}^{T}$, then
(3)$$\left[\begin{array}{c} u\\ v\\ 1 \end{array}\right]=\left[\begin{array}{ccc} 1/d_{x} & 0 & 0\\ 0 & 1/d_{y} & 0\\ 0 & 0 & 1 \end{array}\right]\left[\begin{array}{c} x_{p}\\ y_{p}\\ 1 \end{array}\right]$$

In summary, combining Formulas (1)–(3), the transformation matrix K from the camera coordinate point $P_{c}={\left(x_{c},y_{c},z_{c}\right)}^{T}$ to the pixel coordinate point $Pix={\left(u,v,1\right)}^{T}$ can be obtained:(4)$$K=\left[\begin{array}{ccc} 1/d_{x} & 0 & 0\\ 0 & 1/d_{y} & 0\\ 0 & 0 & 1 \end{array}\right]\left[\begin{array}{ccc} f & 0 & x_{0}\\ 0 & f & y_{0}\\ 0 & 0 & 1 \end{array}\right]=\left[\begin{array}{ccc} f_{x} & 0 & u_{0}\\ 0 & f_{y} & v_{0}\\ 0 & 0 & 1 \end{array}\right]$$

Among them, $f_{x}=f/d_{x}$ and $f_{y}=f/d_{y}$ are called the scale factors of the camera in the u-axis and v-axis directions:(5)$$z_{c}\left[\begin{array}{c} u\\ v\\ 1 \end{array}\right]=K\cdot \left[\begin{array}{cc} R & T\\ 0 & 1 \end{array}\right]\left[\begin{array}{c} x_{w}\\ y_{w}\\ z_{w}\\ 1 \end{array}\right]$$

Equation (Equation 5) represents the transformation from world coordinates to pixel coordinates. The above explanation covers the principles of camera imaging. Building on this foundation, we propose a new depth measurement method.

In conventional monocular camera distance measurement, directly measuring depth is challenging because it lacks stereoscopic information. To address this issue, this study employs an innovative approach, as shown in Figure 7, using a fixed-length marker wristband as a depth-calibration tool. By applying the principles of camera imaging, we can accurately calculate the distance between the camera and the marker wristband. Ultimately, by comparing the known length of the marker with the image captured by the camera, we achieve precise mapping calculations of real-world compression depth.

During the execution of the program, it is necessary to read the detection frame displacement, denoted by $B_{0}$, at the current window resolution. The resolution conversion function f converts the detection frame displacement $B_{0}$ at the current window resolution to the pixel height $B_{p}$ at the ideal camera resolution, i.e.:(6)$$B_{p}=f\left(B_{0}\right)$$

From Figure 6, $B_{p}$ is the vertical displacement of the marker captured by the camera, $L^{\prime }$ is the focal length of the camera, L is the horizontal distance between the marker and the camera, R is half of the vertical displacement of the marker, H is the vertical displacement of the marker, and the following equation is obtained:(7)$$\left\{\begin{array}{c} \tan\left(a\right)=\frac{A_{p}}{2L^{\prime }}\\ \tan\left(b\right)=\frac{B_{p}}{2L^{\prime }}\\ \tan\left(b\right)=\frac{R}{L} \end{array}\right.$$

The following is obtained from Equation (Equation 7):(8)$$\frac{\tan(a)}{\tan(b)}=\frac{A_{p}}{B_{p}}$$

Substituting $\tan\left(b\right)=\frac{R}{L}$ from Equation (Equation 8) yields:(9)$$L=\frac{R\times A_{p}}{B_{p}\times \tan\left(a\right)}$$

In summary:(10)$$H=2\tan\left(b\right)\times L=\frac{B_{p}\times L}{L^{\prime }}$$

H is the realistic depth of compression that we seek.

### 2.4. Edge Device Algorithm Optimization

Given the limited computational power of existing edge devices, a special optimization method is needed to enhance the timeliness of CPR action recognition, which requires high accuracy and real-time processing. As illustrated in Figure 8, the deep-learning algorithm model is first converted into weights compatible with the corresponding NPU. During this conversion process, MMSE algorithms and lossless pruning are employed to obtain more lightweight weights. Next, a multithreading scheme is designed. Two threads on the CPU handle the algorithm’s pre-processing and post-processing, while one thread on the NPU handles the inference phase. The RGA method is applied to image processing during both the pre- and post-processing stages. Finally, NEON instructions are used during the algorithm’s compilation phase.

By using the MMSE algorithm for weight quantization and applying RGA and NEON acceleration, the algorithm’s size is reduced, computational overhead is minimized, and inference speed is increased. Lossless pruning during model quantization effectively prevents accuracy degradation. The multithreading design enables asynchronous processing between the CPU and NPU, significantly improving the model’s performance on edge devices.

## 3. Experiments and Results

### 3.1. Datasets

The dataset used in this study consists of video frames of CPR actions captured by student volunteers from Nanjing University of Posts and Telecommunications in various scenarios. These videos encompass different indoor and outdoor environments and lighting conditions. The environments include objects with colors similar to the marker wristbands. The volunteer group comprised students with and without first aid knowledge to ensure data diversity and broad applicability. Videos are selected based on clarity, shooting angle, and visibility of the marker wristbands. Videos with low image quality due to blurriness, overexposure, or unclear markers are excluded to maintain high quality and consistency in the dataset. The original dataset contains 1479 images, which are augmented to 8874 images. To ensure the model’s robustness and generalization ability, the dataset is divided into training, testing, and validation sets in an 8:1:1 ratio, comprising 7081, 897, and 896 images, respectively. The experiments focus on a single object type, the marker wristband, ensuring the model specifically targeted this object.

### 3.2. Experimental Setting and Evaluation Index

The marker wristband used in the experiments is 33.40 cm long, 3.80 cm wide, and fluorescent green. The experiments are conducted on an NVIDIA GEFORCE RTX 6000 GPU with 24 GB of memory to ensure efficient training. The model is trained without using pre-trained weights. Image processing and data-augmentation techniques are employed to reduce overfitting and improve recognition accuracy. The training parameters are set as follows: image resolution of 352×352, 300 epochs, a learning rate of 0.001, and a batch size of 512. To ensure annotation accuracy and consistency, professionally trained volunteers use the LabelMe tool to annotate images, accurately marking each wristband within the boundary boxes to avoid unnecessary noise. During the training phase, we implement basic image quality control measures, including checking image clarity, brightness, and contrast. All images are cropped and scaled to a uniform 352×352 pixels to standardize the input data format. To enhance the model’s generalization ability and reduce overfitting, various data-augmentation techniques are applied. These included random rotation, horizontal and vertical flipping, random scaling, and slight color transformations (such as hue and saturation adjustments) to simulate different lighting conditions. These steps ensure the dataset’s quality, making the model more robust and reliable. The training process of the dataset is illustrated in Figure 9a, showing batch 0, while Figure 9b shows the testing of batch 0 using the dataset labels.

True positives (TP) refer to the number of instances where the actual condition is “yes” and the model also predicts “yes”. True negatives (TN) refer to the number of instances where the actual condition is “no” and the model correctly predicts “no”. False positives (FP) occur when the model incorrectly predicts “yes” for an actual “no” scenario, leading to false alarms. Conversely, False negatives (FN) occur when the model incorrectly predicts “no” for an actual “yes” scenario [49]. Precision and recall are calculated using Equations (11) and (12), respectively [50,51].
(11)$$\mathrm{Precision}\left(P\right)=\frac{\mathrm{TP}}{\mathrm{TP}+\mathrm{FP}}$$
(12)$$\mathrm{Recall}\left(R\right)=\frac{\mathrm{TP}}{\mathrm{TP}+\mathrm{FN}}$$

### 3.3. OpenPose for CPR Recognition

During the process of performing CPR with an AED device, some errors may be difficult to detect through direct observation by a physician. Therefore, it is necessary to use OpenPose to draw skeletal points. As shown in Figure 10, three common incorrect CPR scenarios are identified: obscured arm movements due to dark clothing, kneeling on one knee, and non-vertical compressions. In the first scenario, dark clothing reduces the contrast with the background, making it difficult to clearly distinguish the edges of the arms. This issue is exacerbated in low-light conditions, making arm movements even more blurred and harder to identify. In the second scenario, kneeling on one knee causes the rescuer’s body to be unstable, affecting the stability and effectiveness of the compressions. In the third scenario, non-vertical compressions cause the force to be dispersed, preventing it from being effectively concentrated on the patient’s chest, thereby affecting the depth and effectiveness of the compressions. These issues can all be addressed using OpenPose. After posture recognition, physicians can remotely provide voice reminders, allowing for the immediate correction of these otherwise difficult-to-detect incorrect postures.

### 3.4. Ablation Experiment

CPR-Detection is an improved object-detection model designed to optimize recognition accuracy and speed. In medical CPR scenarios, due to the limited computational power of edge devices, smaller image inputs (352 × 352 pixels) are typically used to achieve the highest possible mAP0.5. To assess the specific impact of the new method on mAP0.5, ablation experiments are conducted on Yolo-FastestV2. The study independently and jointly tests the effects of the PConv, MLCA, and STD-FPN modules on model performance. The results, as shown in Table 1, clearly demonstrate that these modules, whether applied alone or in combination, enhance the model’s mAP0.5: introducing PConv improves mAP0.5 by 0.44%, optimizing the extraction and representation of positional features [42]. Using MLCA increases mAP0.5 by 0.44%, effectively enhancing the model’s ability to process channel-level and spatial-level information [43]. Applying the STD-FPN structure results in a 0.11% mAP0.5 improvement, optimizing feature fusion and positional enhancement. Combining PConv and MLCA boosts mAP0.5 to 96.87%, achieving a 0.83% increase. The combination of PConv and STD-FPN raises mAP0.5 by 0.95%, better integrating local and global features. The combined use of all three modules increases mAP0.5 by 1.00%, slightly increasing FLOPs but reducing the number of parameters.

These improvements significantly enhance the model’s ability to recognize small targets in CPR scenarios, ensuring higher accuracy while maintaining real-time detection, and demonstrating the superiority of the CPR-Detection model. The combined use of the three modules fully leverages their unique advantages, enabling the model to adapt flexibly to different input sizes and application scenarios, providing an ideal object-detection solution for medical emergency scenarios that demand high accuracy and speed.

### 3.5. Compared with State-of-the-Art Models

To evaluate the impact of the proposed method on the model’s feature-extraction capabilities, the CPR-Detection model is compared with six state-of-the-art lightweight object-detection models, including FastestDet and Yolo-FastestV2 based on the YoloV5 architecture, as well as other official lightweight models. This comparison aims to demonstrate the effectiveness of the new method in improving model performance. Compared to Yolo-FastestV2, the improved CPR-Detection model significantly enhance feature-extraction capabilities. Table 2 presents a quantitative comparison of these models in terms of FLOPs, parameter count, mAP0.5, and mAP0.5:0.95.

As shown in Table 2, the comparison results of CPR-Detection with other models in terms of mAP0.5 are as follows: CPR-Detection’s mAP0.5 improved by 1.02% compared to YoloV7-Tiny; by 6.84% compared to NanoDet-m; by 11.46% compared to FastestDet; and by 1.00% compared to Yolo-FastestV2. Although CPR-Detection’s mAP0.5 is slightly lower than YoloV3-Tiny and YoloV5-Lite (1.45% and 1.16% lower, respectively), it has fewer parameters and lower computational costs compared to these models. This balance strikes an optimal point between speed and accuracy, making it an ideal choice for medical emergency scenarios with limited computational resources.

### 3.6. Measurement Results

One of the key parameters in CPR is the number and frequency of compressions. In this study, we identify each effective compression by analyzing the peaks and troughs of hand movements in the video, with each complete peak–trough cycle representing one compression. The frequency is calculated based on the number of effective compressions occurring per unit of time. Extensive testing shows that the accuracy of compression count and frequency exceeds 98%, with depth accuracy over 90% and errors generally within 1 cm. The errors in count and frequency are mainly due to initial fluctuations of the marker, while depth errors were often caused by inconsistencies in marker performance under different experimental conditions, such as camera angle and lighting changes. The video analysis-based method for measuring CPR compression count, frequency, and depth proposed in this study is highly accurate and practical. It is crucial for guiding first responders in performing standardized CPR, significantly enhancing the effectiveness of emergency care. Although there are some errors, further optimization of the algorithm and improvements in data-collection methods are expected to enhance measurement accuracy.

Figure 11a shows the depth variance distribution for 100 compressions. Most data points have depth errors within ±1 cm, meeting CPR operational standards and demonstrating the high accuracy of the measurement system. However, a few data points exceed a 1 cm depth error, likely due to changes in experimental conditions, such as slight adjustments in camera angle or lighting intensity, which can affect the visual recognition accuracy of the wristband. Figure 11b illustrates the accuracy for each of the 100 measurement tests conducted. A 90% accuracy threshold is set to evaluate the system’s performance. Results indicate that the vast majority of measurements exceed this threshold, confirming the system’s high reliability in most cases. However, there are a few instances where accuracy falls below 90%, highlighting potential weaknesses in the system, such as improper actions, insufficient device calibration, or environmental interference. Future work will focus on diagnosing and addressing these issues to improve the overall performance and reliability of the system.

### 3.7. AED Application for CPR

When using the AED edge device, the user should wear the wristband on their arm and prepare for CPR. The usage process is as follows. After activating the AED edge device, the data-collection unit starts automatically. Once the intelligent emergency function is initiated, the device automatically activates the AI recognition module, capturing real-time images of the emergency scene and collecting data for AI image recognition. During CPR, the AI recognition module uses multiple algorithms to assess whether the procedure meets standards. The voice playback and video display modules provide corrective prompts based on AI processing feedback. The storage module continuously records device operation, emergency events, detection, and AI recognition feedback. Medical emergency personnel can view real-time audio-visual information, location data, AED data, and AI recognition feedback sent by the intelligent module via the emergency platform server. The server also transmits this data back to the device. The intelligent module connects to the emergency platform server through the communication module, retrieves the server’s audio-visual data, and plays it through the voice playback and video display modules. As illustrated in Figure 12, our algorithm’s effectiveness in practical applications is demonstrated. We capture two frames from the AED edge device video after activation, showing the displayed activation time, compression count, frequency, and depth. Additionally, we used OpenPose to visualize skeletal points, capturing the arm’s local motion trajectory during compressions [16]. This helps doctors assess the correctness of the posture via the emergency platform server.

As shown in Figure 13, after optimizing the algorithm on the edge device, the initial frame rate of 8 FPS was significantly improved. By applying quantization methods, the frame rate increased by 5 FPS. Pruning techniques add another 2 FPS, and the asynchronous method contributed an additional 7 FPS. Further enhancements are achieved with RGA and NEON, which improve the frame rate by 1 FPS and 2 FPS, respectively. Overall, the frame rate increases from 8 FPS to 25 FPS, validating the feasibility of these optimization methods.

## 4. Discussion

The application of artificial intelligence in CPR action standardization addresses the limitations of traditional methods. Traditional CPR training relies on classroom simulations, which fail to replicate the stress of actual cardiac arrest events, while VR and AR technologies, though educational, lack real-time application [6,7,8]. Unlike mainstream techniques that have not fully embraced AI, DLCAS pioneers real-time AI interventions on AEDs, offering immediate feedback and corrective actions to improve CPR accuracy and survival rates. By utilizing advanced deep-learning methods like OpenPose, the CPR-Detection algorithm, and edge device optimization, DLCAS achieves high precision in posture detection and compression metrics. Specifically, it boasts a mean average precision (mAP) of 97.04% and impressive accuracy in depth and count measurements. Furthermore, DLCAS is optimized for edge devices, enhancing processing speed from 8 to 25 fps to meet emergency demands.

In the third part of this study, we evaluate the effectiveness of the DLCAS method through a series of experiments. The figures and quantitative performance metrics of the experimental results highlight the superiority of our approach. Qualitatively, Figure 10, Figure 11 and Figure 12 demonstrate significant improvements in our method’s ability to capture arm movements and compression depth accuracy. Additionally, Table 1 and Table 2 present comprehensive quantitative results across these datasets, consistently indicating that our proposed CPR-Detection algorithm outperforms existing models in terms of accuracy and efficiency. Section 3.7 provides a detailed account of how we optimize the algorithm for edge devices to ensure high performance in practical applications.

Our method demonstrates exceptional performance in both quantitative and qualitative experiments, owing to several key innovations. We employ OpenPose for accurate and rapid recognition of human body poses, facilitating physicians’ assessment of posture accuracy via emergency platform servers. In our CPR-Detection approach, we choose PConv over DWSConv to ensure higher efficiency without compromising performance, effectively meeting real-time processing demands. The incorporation of MLCA modules enhances our model’s ability to manage channel-level and spatial-level information. STD-FPN comprehensively integrates shallow and deep features, generating fusion-feature maps rich in positional details that enhance the model’s localization capabilities. Additionally, our depth measurement method guarantees precise mapping of real-world compression depths, while edge-device algorithm optimization ensures efficient performance on edge devices.

The proposed method, while achieving promising results, still has some issues that need to be addressed. Given the strict requirements for data accuracy in medical applications, it is crucial to enhance the accuracy of our model and the stability of the detection boxes in our target-detection algorithm [53]. Additionally, our method relies on the use of marked wristbands, which can consume valuable time in emergency scenarios. In subsequent work, components such as infrared rangefinders will be added to enable distance measurement without the use of a wristband [54]. Reducing the time required for this step would significantly improve the safety of the person being rescued [55].

To address these challenges, future research will focus on several key areas: (1) adopting advanced techniques like dynamic parameter regularization to improve the accuracy and stability of detection boxes by dynamically adjusting regularization parameters throughout the training process [56]; (2) developing markerless motion capture and advanced image-processing algorithms such as infrared rangefinders to eliminate the need for marked wristbands, thereby reducing setup time and increasing the efficiency of emergency interventions [57]; (3) enhancing neural-network interpretability by utilizing techniques such as heat mapping, which will help clinicians better understand and trust AI-assisted decisions [58].

## 5. Conclusions

In this paper, we aim to address the issue related to the lack of standardized cardiopulmonary resuscitation (CPR) actions in automated external defibrillators (AEDs). We propose the deep-learning-based CPR action standardization (DLCAS) method. The first part of DLCAS utilizes OpenPose to identify skeletal points, enabling remote doctors to correct rescuers’ posture through networked AED devices. In the second part of DLCAS, we design the CPR-Detection network. This network uses partial convolution (PConv) to enhance feature representation by focusing on critical spatial information. Additionally, we employ mixed local channel attention (MLCA) on our custom small-target detection-feature pyramid network (STD-FPN). MLCA combines local and global contextual information, improving detection accuracy and efficiency. STD-FPN effectively merges shallow and deep-image features, enhancing the model’s localization capability. Based on CPR-Detection, we introduce a new depth algorithm to measure the rescuers’ compression depth, count and frequency. In the third part of DLCAS, we apply computational optimization methods, including multi-threaded CPU and NPU asynchronous design, RGA, and NEON acceleration, significantly boosting real-time processing efficiency. Extensive experiments on our custom dataset have shown that our method effectively addresses the issue of AED devices’ inability to standardize CPR actions. Furthermore, our method improves the stability and speed of edge devices, validating the applicability of the DLCAS method in current medical scenarios through performance testing.

## Figures and Tables

**Figure 1 sensors-24-04813-f001:**
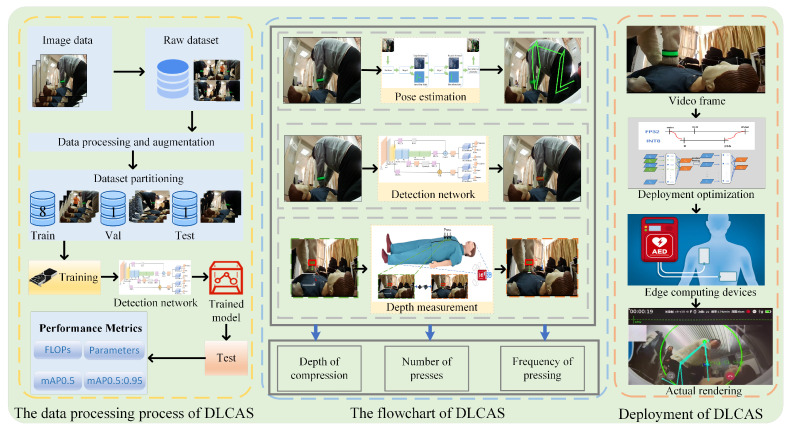
Overall working flowchart.

**Figure 2 sensors-24-04813-f002:**
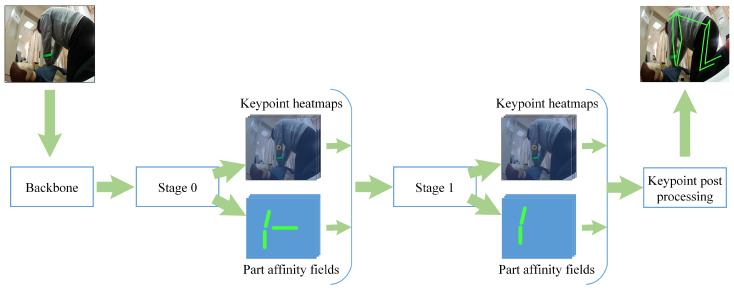
Overall framework of OpenPose.

**Figure 3 sensors-24-04813-f003:**
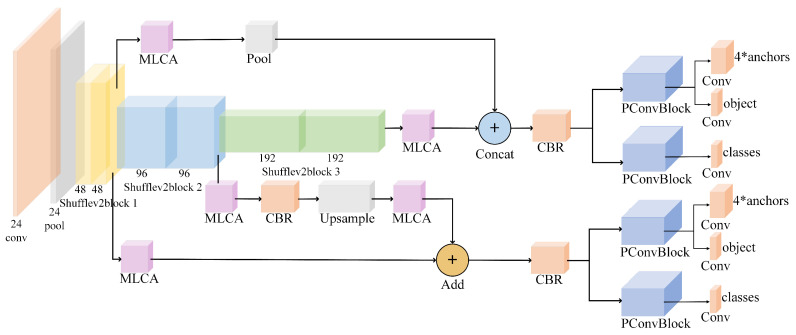
Overall framework of CPR-Detection.

**Figure 4 sensors-24-04813-f004:**
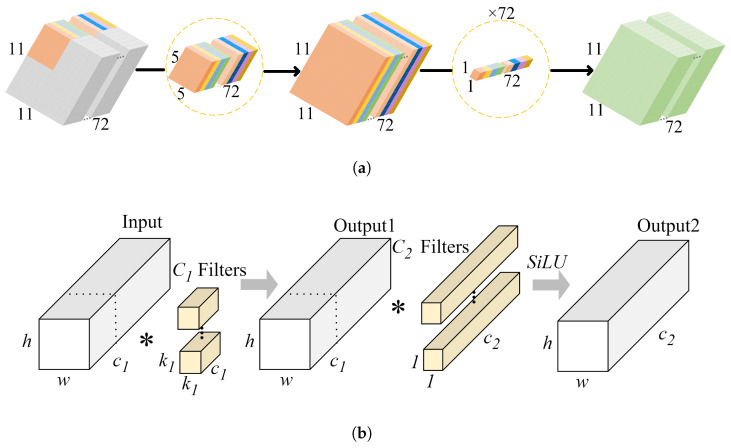
(**a**) DWSConv. (**b**) PConv (The * in the figure indicates convolution calculation).

**Figure 5 sensors-24-04813-f005:**
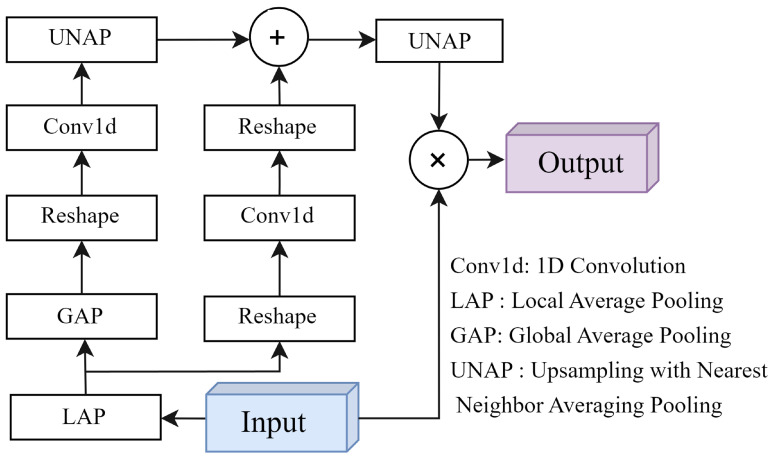
Mixed local channel attention (MLCA).

**Figure 6 sensors-24-04813-f006:**
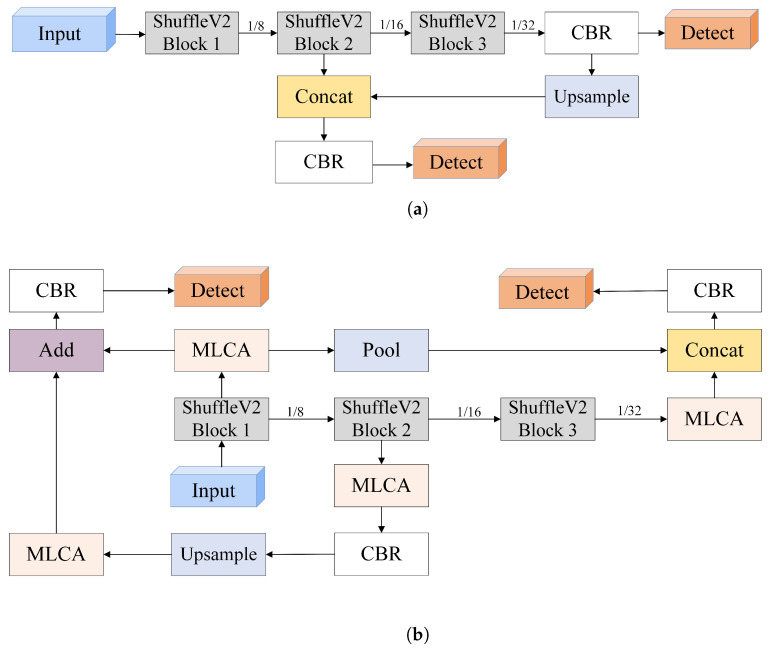
(**a**) The FPN of Yolo-FastestV2. (**b**) Small-target detection-feature pyramid network.

**Figure 7 sensors-24-04813-f007:**
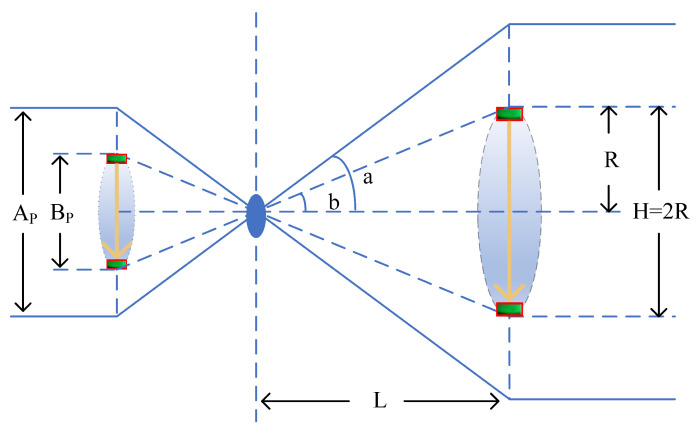
Depth ranging schematic.

**Figure 8 sensors-24-04813-f008:**
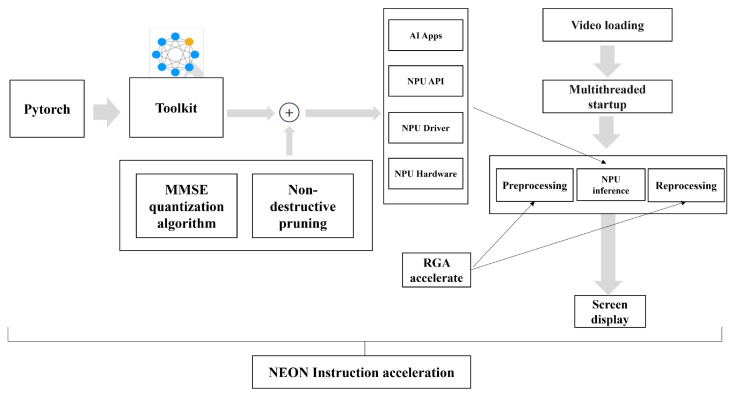
Edge device computing optimization flow chart.

**Figure 9 sensors-24-04813-f009:**
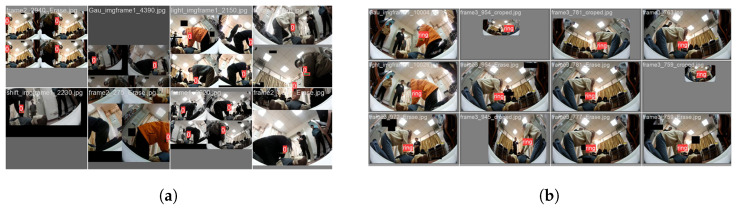
(**a**) Train batch 0 with datasets. (**b**) Test batch 0 labels with datasets.

**Figure 10 sensors-24-04813-f010:**
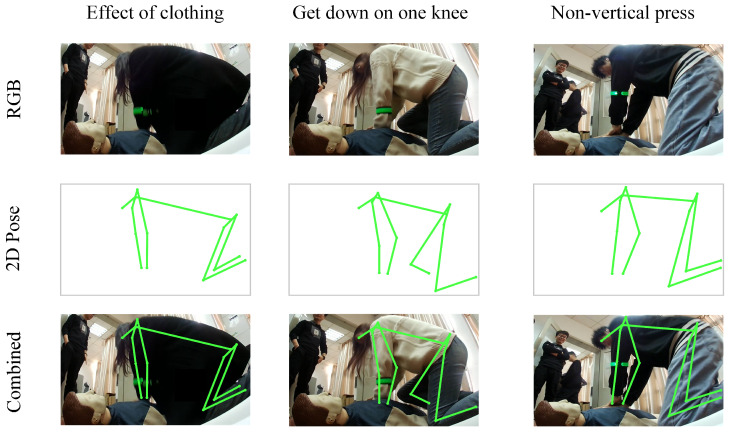
Common incorrect posture images (including RGB, 2D Pose, Combined).

**Figure 11 sensors-24-04813-f011:**
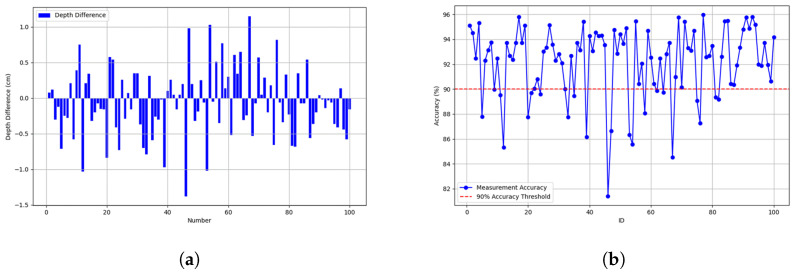
(**a**) Difference between actual depth and measured depth. (**b**) Measurement accuracy over time.

**Figure 12 sensors-24-04813-f012:**
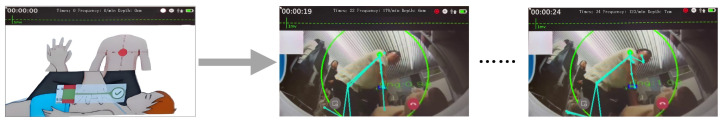
Application scenario flowchart.

**Figure 13 sensors-24-04813-f013:**
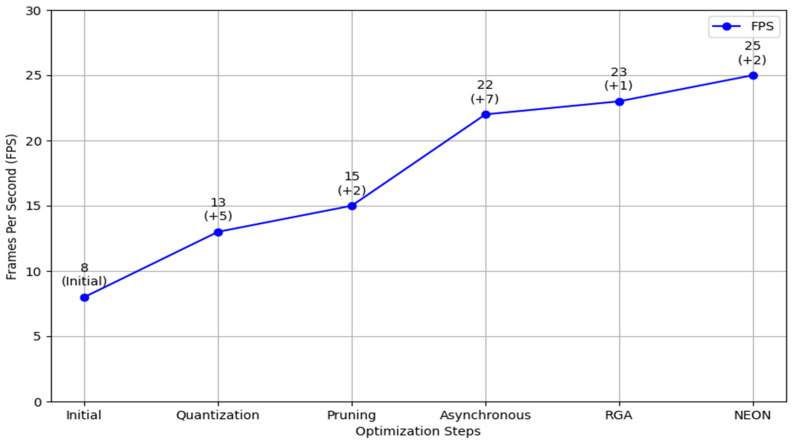
FPS improvement through various optimization steps.

**Table 1 sensors-24-04813-t001:** Validation of the Proposed Method on Yolo-FastestV2.

Index	BASE	PConv	MLCA	STD-FPN	FLOPs	Parameters	mAP0.5	mAP0.5:0.95
1	✓	✗	✗	✗	114.12 K	238.50 K	96.04	72.55
2	✓	✓	✗	✗	105.98 K	213.30 K	96.48	73.89
3	✓	✗	✓	✗	114.36 K	238.52 K	96.48	75.09
4	✓	✗	✗	✓	159.53 K	229.38 K	96.15	71.12
5	✓	✓	✓	✗	131.83 K	204.18 K	96.99	75.16
6	✓	✓	✗	✓	106.22 K	213.32 K	96.87	76.57
**7**	**✓**	**✓**	**✓**	**✓**	**132.15 K**	**204.20 K**	**97.04**	**75.13**

**Table 2 sensors-24-04813-t002:** Model Comparison.

Method	Size	FLOPs	Parameters	mAP0.5	mAP0.5:0.95
YoloV3-Tiny [34]	352 × 352	1.97 G	8.66 M	98.49	80.42
YoloV7-Tiny [52]	352 × 352	13.2 G	6.01 M	96.02	66.05
NanoDet-m [37]	352 × 352	0.87 G	0.96 M	90.20	65.70
Yolo-FastestV2 [40]	352 × 352	0.11 G	0.23 M	96.04	72.55
FastestDet [41]	352 × 352	0.13 G	0.23 M	85.58	52.90
YoloV5-Lite [39]	352 × 352	3.70 G	1.54 M	98.20	77.20
**CPR-Detection**	352 × 352	**0.13 G**	**0.20 M**	**97.04**	**75.13**

## Data Availability

Due to privacy protection for student volunteers, the data supporting the reported results will be made available upon request after acceptance and following privacy protection inquiries.

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
