# Peer review of "A Deep-Learning-Based CPR Action Standardization Method"

_sensors, 2024, doi:10.3390/s24154813_

Round 1
Reviewer 1 Report
Comments and Suggestions for Authors
The authors described that Deep Learning-based CPR Action Standardization (DLCAS) addresses this by using OpenPose to detect posture, a CPR-Detection algorithm to measure compression depth, count, and frequency, and optimizing for edge devices to improve real-time processing. Their custom dataset shows the CPR-Detection algorithm achieves a mAP0.5 of 97.04%, with parameters reduced to 0.20M and FLOPs to 132.15K. Depth measurement accuracy is 90% (±1 cm), and count/frequency accuracy is 98% (±2 counts). Processing speed on edge devices increased from 8fps to 25fps.
The paper is overall well-written, but I have some specific comments.
[Comments]
Rescuers cannot maintain CPR for extended periods due to fatigue. When a rescuer needs to switch during chest compressions, there is a delay in AI measurement due to device setup. The authors need to explain the method for improving this delay.
Page 11, 3.1. dataset
The authors collected 1,479 images. Please indicate the position in which CPR was performed for the data collection. In real-world scenarios, patients suffer from cardio-pulmonary arrest on a bed, floor, or other surfaces. Can this system be applied to various situations where CPR is performed?
Page 11, 3.1. dataset
Is it common that the authors divide training, testing, validation under the ratio of 8:1:1?
Comments on the Quality of English Language
None
Author Response
Responses to the Reviewer’ Comments(Figures included. Please find attached Pdf file)
We would like to thank the reviewer and editor for their careful reading of the manuscript and for their valuable comments. In the revised paper, the revision parts are marked with blue. In the following explanation of revisions, the reviewer’ comments are provided under the heading of “Question”.
Reviewer #1:
Question 1. Rescuers cannot maintain CPR for extended periods due to fatigue. When a rescuer needs to switch during chest compressions, there is a delay in AI measurement due to device setup. The authors need to explain the method for improving this delay.
Answer: Thank you for your valuable feedback and comments. Regarding the AI measurement delay caused by device settings, we have optimized the CPR detection algorithm and NPU to reduce parameters and floating-point operations (FLOPs), thereby enhancing the processing speed on edge devices. As a result, our algorithm's performance on edge devices has improved from 8fps to 25fps, significantly reducing latency.
Potential delays may occur when rescuers need to switch wristbands due to fatigue. To address this issue, we propose the following improvements:
1) Quick Switch Mechanism: Develop a quick switch mechanism and design wristbands that are easier to detach and wear, reducing the time required for switching and thus minimizing the delay.
2) Parallel Monitoring with Multiple Wristbands: Introduce a parallel monitoring system with multiple wristbands to ensure seamless handover when switching, reducing interruptions and delays caused by wristband changes.
We are committed to further exploring and validating the effectiveness of these methods in future research. Your insightful feedback has been instrumental in helping us improve our system.
Thank you once again for your constructive comments.
Question 2. The authors collected 1,479 images. Please indicate the position in which CPR was performed for the data collection. In real-world scenarios, patients suffer from cardio-pulmonary arrest on a bed, floor, or other surfaces. Can this system be applied to various situations where CPR is performed?
Answer: Thank you for your feedback and comments. The most common scenario for sudden cardiac arrest is the patient collapsing on the floor. Therefore, all the images we collected were taken with the patients on the floor and the AED device positioned at the same level as the patient.
As shown in Figure 1, the AED should be placed beside the rescuer, and the camera should be installed on the top of the AED device, at the same level as the rescued person.
In real-world scenarios, whether the patient is on a bed, floor, or any other surface, as long as the AED device is at the same level as the patient, our system can function properly. This conclusion is based on extensive testing in various conditions.
|
|
|
Figure 1. Scenarios for using AED.
Question 3. Is it common that the authors divide training, testing, validation under the ratio of 8:1:1?
Answer: Thank you for your comments. Splitting the dataset into an 8:1:1 ratio for training, validation, and testing is indeed a common practice, especially in the fields of deep learning and machine learning. This method ensures that the model has sufficient data for learning while also providing enough data for validation and testing to evaluate the model's performance. There is no universally best split percentage, but the 8:1:1 ratio is the optimal split for our dataset and model requirements.

Reviewer 2 Report
Comments and Suggestions for Authors
I read with interest the manuscript entitled "DLCAS: A Deep Learning-Based CPR Action Standardization Method"
Do not use abbreviations in the title.
Please state the full name of the word when the abbreviation first appears in the text.
Use MeSH terms for keywords.
When you mention the shortcomings in the existing CPR methods in the introduction, please state them clearly. Also, mention the latest CPR algorithms.
In the introduction, you have already touched on a number of algorithms that are more appropriate for the methods section. I would like the introduction to be shorter and more concise, citing relevant references without going into deeper explanations.
Please remove the paragraph "The following is an outline of this study. Section 2 discusses the modules and algorithms we use. In Section 3, the introduction of the data set and the introduction of data preprocessing are carried out. Section 4 provides the results of the experiments. Section 5 discusses our conclusions."
Parts of the text, such as "In this section, we first introduce the principles of OpenPose, followed by the design details of CPR-Detection. Next, we explain the depth measurement scheme based on object detection algorithms. Finally, we discuss the optimization of computational methods for edge devices." please remove. It is clear to readers from the subtitles what will be discussed in which section.
Within the results section, you have not shown the exact quantitative results. Please present all results in detail.
Within the discussion section, part of the text is certainly more appropriate for the results section. At the beginning of the discussion, you must briefly present the most relevant results, which you must then comment on in relation to the previous knowledge on the given topic.
Also, within the discussion you must detail all the strengths and limitations of your study, of which there are many. Please deal with this topic in detail.
You don't have a single reference listed within the discussion section, which is unacceptable. It is clear that the focus of the discussion must be on the strength and limitations of the study in relation to previous research on the mentioned topic.
It is sufficient to summarize the conclusion in the 3-4 most relevant sentences in relation to the set aims of the study at the end of the introduction.
In conclusion, there are many more references of interest on the above topic that you did not include in your manuscript. In general, the topic is of interest to readers, but I ask that you thoroughly revise the structure of the manuscript in accordance with the instructions.
Comments on the Quality of English LanguageModerate editing of English language required.
Author Response
Responses to the Reviewer’ Comments
We would like to thank the reviewer and editor for their careful reading of the manuscript and for their valuable comments. In the revised paper, the revision parts are marked with blue. In the following explanation of revisions, the reviewer’ comments are provided under the heading of “Question”.
Reviewer #2:
Question 1. I read with interest the manuscript entitled "DLCAS: A Deep Learning-Based CPR Action Standardization Method"
Do not use abbreviations in the title.
Please state the full name of the word when the abbreviation first appears in the text.
Answer: Thank you for your insightful comments and suggestions on our manuscript. We have addressed your feedback and made the necessary changes as follows:
1) We have revised the title of our manuscript to "A Deep Learning-Based CPR Action Standardization Method," ensuring that no abbreviations are used in the title.
2) We have stated the full name of each abbreviation the first time it appears in the text, providing clarity and improving readability.
We appreciate your thorough review and believe these changes have strengthened our manuscript.
Thank you once again for your valuable feedback.
Question 2. Use MeSH terms for keywords.
Answer: Thank you for your valuable feedback on our manuscript. We have carefully reviewed your suggestions and made the necessary revisions to the keywords. The original keywords "CPR; AED; depth measurement; object detection; OpenPose; edge computing" have been updated to align with MeSH terms as follows:keywords "Deep Learning; Processing Speed; Cardiopulmonary Resuscitation; Defibrillators; Reference Standards; Posture"
We believe these changes enhance the clarity and relevance of our manuscript. We appreciate your guidance in improving the quality of our work.
Thank you for your consideration.
Question 3. When you mention the shortcomings in the existing CPR methods in the introduction, please state them clearly. Also, mention the latest CPR algorithms.
Answer: Thank you for your comments. Regarding the shortcomings in the existing CPR methods mentioned in the introduction, we have now clearly stated them and also included the latest CPR algorithms.
Original:
Line 186-187: Research has highlighted the limitations of traditional CPR training methods and the potential of AI to transform CPR training and execution. ……This approach represents a significant advancement over traditional methods, which lack the capability to dynamically adapt and guide rescuers in real-time.
Modified:
Line 186-187: Current CPR methods have several limitations, particularly in their effectiveness during real emergency situations. Traditional CPR training relies heavily on classroom simulations, which cannot replicate the pressure and urgency of actual cardiac arrest scenarios. This can lead to improper performance during real emergencies [6]. Although VR and AR technologies are being used to enhance CPR training, they remain primarily educational tools and are not widely integrated into real-time emergency applications [7,8]. Moreover, mainstream CPR techniques have not fully incorporated AI assistance; advancements have focused more on mechanical devices and VR/AR training rather than real-time AI intervention [8, 9]. Recent advancements in CPR algorithms have started to address these issues by integrating computational models and machine learning techniques. For instance, the use of integrated computational models of the cardiopulmonary system to evaluate current CPR guidelines has shown potential in improving CPR effectiveness[10]. Additionally, machine learning has been used to identify higher survival rates during extracorporeal cardiopulmonary resuscitation, significantly enhancing survival outcomes [11]. The future trends in CPR technology indicate that the combination of AI and machine learning will continue to evolve, potentially predicting and shaping technological innovations in this field [12]. To bridge the gap between training and real-time application, this paper proposes the first application of posture estimation and object detection algorithms on AEDs to assist in real-time CPR action standardization, extending their use to actual emergency rescues. This innovative approach addresses the lack of real-time AI-assisted intervention in current CPR methods, thereby improving the accuracy and effectiveness of lifesaving measures during OHCA incidents. By integrating AI technology into AED devices, we aim to provide immediate feedback and corrective actions during CPR, potentially increasing survival rates and reducing risks associated with improper CPR techniques. This approach represents a significant advancement over traditional methods, which lack the ability to dynamically adjust in real-time and guide rescuers [13,14].
Reference:
[6] Rodríguez-Matesanz, M.; Guzmán-García, C.; Oropesa, I.; Rubio-Bolivar, J.; Quintana-Díaz, M.; Sánchez-González, P. A New Immersive virtual reality station for cardiopulmonary resuscitation objective structured clinical exam evaluation. Sensors 2022,22, 4913.
[7] Krasteva, V.; Didon, J.P.; Ménétré, S.; Jekova, I. Deep Learning Strategy for Sliding ECG Analysis during Cardiopulmonary Resuscitation: Influence of the Hands-Off Time on Accuracy. Sensors 2023, 23, 4500.
[8] Xie, J.; Wu, Q. Design and Evaluation of CPR Emergency Equipment for Non-Professionals. Sensors 2023, 23, 5948.
[9] Tang, X.; Wang, Y.; Ma, H.; Wang, A.; Zhou, Y.; Li, S.; Pei, R.; Cui, H.; Peng, Y.; Piao, M. Detection and Evaluation for High-Quality Cardiopulmonary Resuscitation Based on a Three-Dimensional Motion Capture System: A Feasibility Study. Sensors 2024,
24, 2154.
[10] Daudre-Vignier, C.; Bates, D.G.; Scott, T.E.; Hardman, J.G.; Laviola, M. Evaluating current guidelines for cardiopulmonary resuscitation using an integrated computational model of the cardiopulmonary system. Resuscitation 2023, 186, 109758.Version July 5, 2024 submitted to Journal Not Specified 18 of 19
[11] Crespo-Diaz, R.; Wolfson, J.; Yannopoulos, D.; Bartos, J.A. Machine learning identifies higher survival profile in extracorporeal cardiopulmonary resuscitation. Critical Care Medicine 2024, 52, 1065–1076.
[12] Semeraro, F.; Schnaubelt, S.; Hansen, C.M.; Bignami, E.G.; Piazza, O.; Monsieurs, K.G. Cardiac arrest and cardiopulmonary resuscitation in the next decade: Predicting and shaping the impact of technological innovations. Resuscitation 2024, 200, 110250.
[13] Shrimpton, A.J.; Brown, V.; Vassallo, J.; Nolan, J.; Soar, J.; Hamilton, F.; Cook, T.; Bzdek, B.R.; Reid, J.P.; Makepeace, C.; et al. A quantitative evaluation of aerosol generation during cardiopulmonary resuscitation. Anaesthesia 2024, 79, 156–167.
[14] Kao, C.L.; Tsou, J.Y.; Hong, M.Y.; Chang, C.J.; Tu, Y.F.; Huang, S.P.; Su, F.C.; Chi, C.H. A novel CPR-assist device vs. established chest compression techniques in infant CPR: A manikin study. The American Journal of Emergency Medicine 2024, 77, 81–86.
Question 4. In the introduction, you have already touched on a number of algorithms that are more appropriate for the methods section. I would like the introduction to be shorter and more concise, citing relevant references without going into deeper explanations.
Answer: Thank you for your insightful comments. We have revised the introduction to be shorter and more concise by simplifying the discussion of algorithms.
Original:
Line 192-193: In recent years, ……However, for our dataset, FastestDet underperformed mainly due to its single detection head design limiting the utilization of features with different receptive fields and lacking sufficient feature fusion, resulting in insufficient accuracy in locating small objects.
Modified:
Line 192-193: In recent years, various lightweight object detection networks have been proposed and widely applied in traffic management [23–26], fire warning systems [27], anomaly detection [28–30], and facial recognition [31–33]. Redmon et al. [34] introduced an end-to-end object detection model using Darknet53, incorporating k-means clustering for anchor boxes, multi-label classification for class probabilities, and a feature pyramid network for multi-scale bounding box prediction. Wong et al. [35] developed Yolo Nano, a compact network for embedded object detection with a model size of approximately 4.0MB. Hu et al. [36] improved the Yolov3-tiny network by using depthwise distributed convolutions and squeeze-and-excitation blocks, creating Micro-Yolo to reduce parameters and optimize performance. Lyu [37] proposed NanoDet, an anchor-free model using generalized focal loss and GhostPAN for enhanced feature fusion, increasing accuracy on the COCO dataset by 7% mAP. Ge et al. [38] modified Yolo to an anchor-free mode with a decoupled head and SimOTA strategy, significantly enhancing performance. For example, Yolo Nano achieved 25.3% AP on the COCO dataset with only 0.91M parameters and 1.08G FLOPs, surpassing NanoDet by 1.8%, while the improved Yolov3 AP increased to 47.3%, exceeding the current best practice by 3.0%. Yolov5 Lite [39] optimized inference speed by adding shuffle channels and pruning head channels while maintaining high accuracy. Dogqiuqiu [40] developed the Yolo-Fastest series for single-core real-time inference, reducing CPU usage. Yolo-FastestV2 used the ShufflenetV2 backbone, decoupled the detection head, reduced parameters, and improved the anchor matching mechanism. Dogqiuqiu [41] further proposed FastestDet, simplifying to a single detection head, transitioning to anchor-free, and increasing candidate objects across grids for ARM platforms. However, for our dataset, FastestDet underperformed mainly due to its single detection head design, limiting the utilization of features with different receptive fields and lacking sufficient feature fusion, resulting in insufficient accuracy in locating small objects.
Reference:
[23] Wei, Y.; Zhao, L.; Zheng, W.; Zhu, Z.; Zhou, J.; Lu, J. SurroundOcc: Multi-Camera 3D Occupancy Prediction for Autonomous Driving. In Proceedings of the 2023 IEEE/CVF International Conference on Computer Vision (ICCV), 2023.
[24] Wu, D.; Liao, M.W.; Zhang, W.T.; Wang, X.G.; Bai, X.; Cheng, W.Q.; Liu, W.Y. Correction to: YOLOP: You Only Look Once for Panoptic Driving Perception. Machine Intelligence Research 2023, p. 952.
[25] Xu, M.; Wang, X.; Zhang, S.; Wan, R.; Zhao, F. Detection algorithm of aerial vehicle target based on improved YOLOv3. Journal of Physics: Conference Series 2022, p. 012022.
[26] Jamiya, S.S.; Rani, P.E. An Efficient Method for Moving Vehicle Detection in Real-Time Video Surveillance. In Proceedings of the Advances in Smart System Technologies, 2020.
[27] Wu, S.; Zhang, L. Using Popular Object Detection Methods for Real Time Forest Fire Detection. In Proceedings of the 2018 11th International Symposium on Computational Intelligence and Design (ISCID), 2018.
[28] Mishra, S.; Jabin, S. Anomaly detection in surveillance videos using deep autoencoder. International Journal of Information Technology (Singapore) 2024, pp. 1111–1122.
[29] Ali, M.M. Real-time video anomaly detection for smart surveillance. IET Image Processing (Wiley-Blackwell) 2023, pp. 1375–1388.
[30] Sun, S.; Xu, Z. Large kernel convolution YOLO for ship detection in surveillance video. Mathematical Biosciences and Engineering 2023, pp. 15018–15043.
[31] Zhang, X.; Xuan, C.; Xue, J.; Chen, B.; Ma, Y. LSR-YOLO: A High-Precision, Lightweight Model for Sheep Face Recognition on the Mobile End. Animals 2023, p. 1824.
[32] Yu, F.; Zhang, G.; Zhao, F.; Wang, X.; Liu, H.; Lin, P.; Chen, Y. Improved YOLO-v5 model for boosting face mask recognition accuracy on heterogeneous IoT computing platforms. Internet of Things 2023, p. 100881.
[33] Sun, F. Face Recognition Analysis Based on the YOLO Algorithm. In Proceedings of the The 4th International Conference on Computing and Data Science (CONF-CDS 2022), 2022.
[34] Redmon, J.; Farhadi, A. YOLOv3: An Incremental Improvement 2018.
[35] Wong, A.; Famuori, M.; Shafiee, M.J.; Li, F.; Chwyl, B.; Chung, J. YOLO Nano: a Highly Compact You Only Look Once Convolutional Neural Network for Object Detection 2019.
[36] Hu, L.; Li, Y. Micro-YOLO: Exploring Efficient Methods to Compress CNN based Object Detection Model. In Proceedings of the International Conference on Agents and Artificial Intelligence, 2021.
[37] Lyu, R. Nanodet-plus: Super fast and high accuracy lightweight anchor-free object detection model. URL: https://github.com/RangiLyu/nanodet 2021.
[38] Ge, Z.; Liu, S.; Wang, F.; Li, Z.; Sun, J. YOLOX: Exceeding YOLO Series in 2021 2021.
[39] Jocher, G.; Nishimura, K.; Mineeva, T.; Vilarino, R. yolov5. Code repository 2020, p. 9.
[40] DOG-QIUQIU, A. dog-qiuqiu/Yolo-Fastest: Yolo-fastest-v1. 1.0 2021.
[41] Ma, X. Fastestdet: Ultra lightweight anchor-free realtime object detection algorithm, 2022.
Question 5. Please remove the paragraph "The following is an outline of this study. Section 2 discusses the modules and algorithms we use. In Section 3, the introduction of the data set and the introduction of data preprocessing are carried out. Section 4 provides the results of the experiments. Section 5 discusses our conclusions."
Parts of the text, such as "In this section, we first introduce the principles of OpenPose, followed by the design details of CPR-Detection. Next, we explain the depth measurement scheme based on object detection algorithms. Finally, we discuss the optimization of computational methods for edge devices." please remove. It is clear to readers from the subtitles what will be discussed in which section.
Answer: Thank you very much for your valuable feedback. I have carefully reviewed your comments and made the necessary revisions to improve the manuscript. Specifically, I have removed the paragraph:
"The following is an outline of this study. Section 2 discusses the modules and algorithms we use. In Section 3, the introduction of the data set and the introduction of data preprocessing are carried out. Section 4 provides the results of the experiments. Section 5 discusses our conclusions."
Thank you again for your insightful suggestions, which have significantly enhanced the clarity and conciseness of the manuscript.
Question 6. Within the results section, you have not shown the exact quantitative results. Please present all results in detail. Within the discussion section, part of the text is certainly more appropriate for the results section. At the beginning of the discussion, you must briefly present the most relevant results, which you must then comment on in relation to the previous knowledge on the given topic.
Answer: Thank you for your constructive feedback. We appreciate your detailed review and recommendations.
Results Section: We will present all quantitative results in detail to provide a clear and comprehensive understanding of our findings. This will include precise numerical data and relevant statistical analyses to support our conclusions.
Discussion Section: We will revise the discussion section to ensure that it focuses on interpreting the results rather than presenting them. At the beginning of the discussion, we will briefly summarize the most relevant results. These results will then be commented on in relation to existing knowledge and previous studies on the topic to provide a thorough comparison and analysis.
Here is the table of contents for my results section:
- Experiments and Results
3.1. Datasets
3.2. Experimental Setting and Evaluation Index
3.3. OpenPose for CPR Recognition
3.4. Ablation Experiment
3.5. Compared With State-of-the-Art Models
3.6. Measurement Results
3.7. AED Application for CPR
Question 7. Also, within the discussion you must detail all the strengths and limitations of your study, of which there are many. Please deal with this topic in detail.
You don't have a single reference listed within the discussion section, which is unacceptable. It is clear that the focus of the discussion must be on the strength and limitations of the study in relation to previous research on the mentioned topic.
Answer: Thank you for your comments, I have detailed all the strengths and limitations of my study and added relevant references. The focus has been on the strengths and limitations of the study in relation to previous studies on the mentioned topic. Here is my modified discussion section:
Line 492-503: AI in CPR action standardization is a crucial technology for improving emergency medical interventions. It offers real-time, high-quality guidance. AI-based CPR methods have increasingly replaced traditional training methods due to their superior performance, becoming a major research focus. To further improve the accuracy and effectiveness of CPR, this paper proposes the DLCAS method. This method integrates pose estimation, the CPR-Detection algorithm, and edge device optimization, providing immediate feedback and corrective actions during CPR. It addresses several critical challenges in CPR action standardization……
[53] Ahmed, S.F.; Alam, M.S.B.; Afrin, S.; Rafa, S.J.; Rafa, N.; Gandomi, A.H. Insights into Internet of Medical Things (IoMT): Data fusion, security issues and potential solutions. Information Fusion 2024, 102, 102060.
[54] Choi, Y.; Park, J.H.; Jeong, J.; Kim, Y.J.; Song, K.J.; Shin, S.D. Extracorporeal cardiopulmonary resuscitation for adult out-of-hospital cardiac arrest patients: time-dependent propensity score-sequential matching analysis from a nationwide population-based registry. Critical Care 2023, 27, 87.
[55] Pu, J.C.; Chen, Y. Data-driven forward-inverse problems for Yajima–Oikawa system using deep learning with parameter regularization. Communications in Nonlinear Science and Numerical Simulation 2023, 118, 107051.
[56] Tian, Z.; Weng, D.; Fang, H.; Shen, T.; Zhang, W. Robust facial marker tracking based on a synthetic analysis of optical flows and the YOLO network. The Visual Computer 2024, 40, 2471–2489.
[57] Wang, Z.; Zhou, Y.; Han, M.; Guo, Y. Interpreting convolutional neural network by joint evaluation of multiple feature maps and an improved NSGA-II algorithm. Expert Systems with Applications 2024, p. 124489.
Question 8. It is sufficient to summarize the conclusion in the 3-4 most relevant sentences in relation to the set aims of the study at the end of the introduction.
Answer: Thank you for your valuable feedback. Based on your suggestions, we have summarized the conclusion into the 3-4 most relevant sentences related to the study's aims and placed it at the end of the introduction. We look forward to your further feedback and guidance.
Original:
Line 194-195: This paper proposes a standard detection method for CPR actions based on AED, utilizing skeletal points to assist in posture estimation. The method identifies the rescuer's marked wristband to measure compression depth, frequency, and count. Considering the limitations of object detection networks on edge devices, we developed the CPR-Detection algorithm based on Yolo-FastestV2. This algorithm not only improves detection accuracy but also simplifies the model structure. Building on this algorithm, we designed a novel compression depth calculation method, which maps actual depth by analyzing the wristband's displacement. We also optimized the network to enhance speed and accuracy on edge devices, ensuring precise compression depth measurement to protect the safety of the rescued individual. Furthermore, we optimized the computation for edge devices.
Modified:
Line 194-195: This paper proposes a standardized CPR action detection method based on AED, utilizing skeletal points to assist in posture estimation. We developed the CPR-Detection algorithm based on Yolo-FastestV2, which includes a novel compression depth calculation method that maps actual depth by analyzing the wristband's displacement. Additionally, we optimized the computation for edge devices to enhance their speed and accuracy.
Question 9. Moderate editing of English language required.
Answer: Thank you for your valuable feedback. We have taken your suggestions seriously and have made extensive revisions to improve the English language in our manuscript. Specifically, we have addressed issues related to tense and grammar to enhance readability and clarity.
We appreciate your detailed review and look forward to your further comments.

Round 2
Reviewer 2 Report
Comments and Suggestions for Authors
I am satisfied with the explanations and answers, but please elaborate and expand the discussion section, which needs to be more comprehensive with many more references to the given topic.
Comments on the Quality of English LanguageModerate editing of English language required.
Author Response
Responses to the Reviewer’ Comments
We would like to thank the reviewer and editor for their careful reading of the manuscript and for their valuable comments. In the revised paper, the revision parts are marked with blue. In the following explanation of revisions, the reviewer’ comments are provided under the heading of “Question”.
Reviewer #2:
Question 3. I am satisfied with the explanations and answers, but please elaborate and expand the discussion section, which needs to be more comprehensive with many more references to the given topic.
Answer: Thank you for your comments. We have now expanded the Discussion section overall with more references on given topics.
Original:
Line 492-503: AI in CPR action standardization is a crucial technology for improving emergency medical interventions. It offers real-time, high-quality guidance. AI-based CPR methods have increasingly replaced traditional training methods due to their superior performance, becoming a major research focus. To further improve the accuracy and effectiveness of CPR, this paper proposes the DLCAS method. This method integrates pose estimation, the CPR-Detection algorithm, and edge device optimization, providing immediate feedback and corrective actions during CPR. It addresses several critical challenges in CPR action standardization……
Modified:
Line 500-509: The application of artificial intelligence in CPR action standardization addresses the limitations of traditional methods. Traditional CPR training relies on classroom simulations, which fail to replicate the stress of actual cardiac arrest events, while VR and AR technologies, though educational, lack real-time application \cite{r49,r50,r51}. Unlike mainstream techniques that have not fully embraced AI, DLCAS pioneers real-time AI interventions on AEDs, offering immediate feedback and corrective actions to improve CPR accuracy and survival rates. By utilizing advanced deep learning methods like OpenPose, the CPR-Detection algorithm, and edge device optimization, DLCAS achieves high precision in posture detection and compression metrics. Specifically, it boasts a mean Average Precision (mAP) of 97.04\% and impressive accuracy in depth and count measurements. Furthermore, DLCAS is optimized for edge devices, enhancing processing speed from 8 to 25 fps to meet emergency demands.
In the third part of this study, we evaluate the effectiveness of the DLCAS method through a series of experiments. The figures and quantitative performance metrics of the experimental results highlight the superiority of our approach. Qualitatively, Figures 10 to 12 demonstrate significant improvements in our method's ability to capture arm movements and compression depth accuracy. Additionally, Tables 1 and 2 present comprehensive quantitative results across these datasets, consistently indicating that our proposed CPR-Detection algorithm outperforms existing models in terms of accuracy and efficiency. Section 3.7 provides a detailed account of how we optimize the algorithm for edge devices to ensure high performance in practical applications.
Our method demonstrates exceptional performance in both quantitative and qualitative experiments, owing to several key innovations. We employ OpenPose for accurate and rapid recognition of human body poses, facilitating physicians' assessment of posture accuracy via emergency platform servers. In our CPR-Detection approach, we choose PConv over DWSConv to ensure higher efficiency without compromising performance, effectively meeting real-time processing demands. The incorporation of MLCA modules enhances our model's ability to manage channel-level and spatial-level information. STD-FPN comprehensively integrates shallow and deep features, generating fusion feature maps rich in positional details that enhance the model's localization capabilities. Additionally, our Depth Measurement Method guarantees precise mapping of real-world compression depths, while Edge Device Algorithm Optimization ensures efficient performance on edge devices.
The proposed method, while achieving promising results, still has some issues that need to be addressed. Given the strict requirements for data accuracy in medical applications, it is crucial to enhance the accuracy of our model and the stability of the detection boxes in our target detection algorithm \cite{r63}. Additionally, our method relies on the use of marked wristbands, which can consume valuable time in emergency scenarios. In subsequent work, components such as infrared rangefinders will be added to enable distance measurement without the use of a wristband \cite{r64}. Reducing the time required for this step would significantly improve the safety of the person being rescued \cite{r59}.
To address these challenges, future research will focus on several key areas: (1) adopting advanced techniques like dynamic parameter regularization to improve the accuracy and stability of detection boxes by dynamically adjusting regularization parameters throughout the training process \cite{r60}; (2) developing markerless motion capture and advanced image processing algorithms such as infrared rangefinders to eliminate the need for marked wristbands, thereby reducing setup time and increasing the efficiency of emergency interventions \cite{r61}; (3) enhancing neural network interpretability by utilizing techniques such as heat mapping, which will help clinicians better understand and trust AI-assisted decisions \cite{r62}.
Reference:
- Rodríguez-Matesanz, M.; Guzmán-García, C.; Oropesa, I.; Rubio-Bolivar, J.; Quintana-Díaz, M.; Sánchez-González, P. A New Immersive virtual reality station for cardiopulmonary resuscitation objective structured clinical exam evaluation. Sensors 2022, 22, 4913.
7. Krasteva, V.; Didon, J.P.; Ménétré, S.; Jekova, I. Deep Learning Strategy for Sliding ECG Analysis during Cardiopulmonary Resuscitation: Influence of the Hands-Off Time on Accuracy. Sensors 2023, 23, 4500.
8. Xie, J.; Wu, Q. Design and Evaluation of CPR Emergency Equipment for Non-Professionals. Sensors 2023, 23, 5948. - 53. Ahmed, S.F.; Alam, M.S.B.; Afrin, S.; Rafa, S.J.; Rafa, N.; Gandomi, A.H. Insights into Internet of Medical Things (IoMT): Data fusion, security issues and potential solutions. Information Fusion 2024, 102, 102060.
Kim, D.; Kang, J.; Na, K.S.; et al. Development of smart glasses monitoring viewing distance using an infrared distance measurement sensor. Investigative Ophthalmology & Visual Science 2024, 65, 2754–2754.
55. Choi, Y.; Park, J.H.; Jeong, J.; Kim, Y.J.; Song, K.J.; Shin, S.D. Extracorporeal cardiopulmonary resuscitation for adult out-of-hospital cardiac arrest patients: time-dependent propensity score-sequential matching analysis from a nationwide population-based
registry. Critical Care 2023, 27, 87.
56. Pu, J.C.; Chen, Y. Data-driven forward-inverse problems for Yajima–Oikawa system using deep learning with parameter regularization. Communications in Nonlinear Science and Numerical Simulation 2023, 118, 107051.
57. Tian, Z.; Weng, D.; Fang, H.; Shen, T.; Zhang, W. Robust facial marker tracking based on a synthetic analysis of optical flows and the YOLO network. The Visual Computer 2024, 40, 2471–2489.
58. Wang, Z.; Zhou, Y.; Han, M.; Guo, Y. Interpreting convolutional neural network by joint evaluation of multiple feature maps and an improved NSGA-II algorithm. Expert Systems with Applications 2024, p. 124489.
